# Enhanced Access to the Health-Related Skin Metabolome by Fast, Reproducible and Non-Invasive WET PREP Sampling

**DOI:** 10.3390/metabo11070415

**Published:** 2021-06-24

**Authors:** Jamie Afghani, Claudia Huelpuesch, Philippe Schmitt-Kopplin, Claudia Traidl-Hoffmann, Matthias Reiger, Constanze Mueller

**Affiliations:** 1Department of Environmental Medicine, Medical Faculty, University Augsburg, D-86156 Augsburg, Germany; jamie.afghani@tum.de (J.A.); claudia.huelpuesch@tum.de (C.H.); claudia.traidl-hoffmann@tum.de (C.T.-H.); 2Chair of Analytical Food Chemistry, School of Life Sciences, Technical University of Munich, 85354 Freising, Germany; Schmitt-kopplin@helmholtz-muenchen.de; 3ZIEL–Institute for Food and Health, Technical University of Munich, 85354 Freising, Germany; 4Christine Kühne Center for Allergy Research and Education (CK-CARE), 7265 Davos, Switzerland; 5German Research Center for Environmental Health, Institute of Environmental Medicine, Helmholtz Zentrum München, D-85764 Neuherberg, Germany; 6Research Unit Analytical BioGeoChemistry, Helmholtz Zentrum München, German Research Center for Environmental Health, D-85764 Neuherberg, Germany

**Keywords:** skin, sampling method, LC-MS^2^, metabolome, non-invasive sampling, skin disease, inter-organ crosstalk

## Abstract

Our skin influences our physical and mental health, and its chemical composition can reflect environmental and disease conditions. Therefore, through sampling the skin metabolome, we can provide a promising window into the mechanisms of the body. However, the broad application of skin metabolomics has recently been hampered by a lack of easy and widely applicable sampling methods. Here, we present a novel rapid, simple, and, most importantly, painless and non-invasive sampling technique suitable for clinical studies of fragile or weakened skin. The method is called WET PREP and is simply a lavage of the skin which focuses on capturing the metabolome. We systematically evaluate WET PREPs in comparison with the non-invasive method of choice in skin metabolomics, swab collection, using ultra-performance liquid chromatography coupled to mass spectrometry (UPLC-MS^2^) on two complementary chromatographic columns (C18 reversed phase and hydrophilic interaction chromatography). We also integrate targeted analyses of key metabolites of skin relevance. Overall, WET PREP provides a strikingly more stable shared metabolome across sampled individuals, while also being able to capture unique individual metabolites with a high consistency in intra-individual reproducibility. With the exception of (phospho-)lipidomic studies, we recommend WET PREPs as the preferred skin metabolome sampling technique due to the quick preparation time, low cost, and gentleness for the patient.

## 1. Introduction

The skin is our largest organ and has a tremendous impact on both our physical and mental health [1]. Through descriptions of its chemical composition, such as the metabolome of the skin, we can begin to understand mechanisms in skin physiology, and eventually reach a point at which we can use important endogenous metabolites to modulate the skin phenotype or use the skin metabolome as a diagnostic tool for systemic diseases. The metabolome defines the entirety of low molecular weight compounds, which includes secreted, exogenous, and endogenous substances. In particular, within the skin, numerous low molecular weight metabolites can be excreted by specialized glands, which promise specific novel biomarkers that are of interest for disease and cosmetic applications.

The metabolome of the skin thereby reflects dynamic changes that occur in relation to the macro and microenvironment, intrinsic factors, and also in relation to skin and systemic disease [2,3,4,5]. In this context, aging is one intrinsic factor that influences the skin metabolome. When comparing older versus younger skin, not only did coenzyme Q10, an anti-aging metabolite, levels change, but also 56 other metabolites differed, including dehydroepiandrosterone-(DHEA-) sulfate and several amino acids [3]. The physical macroenvironment can be reflected by the metabolome as well. UV light exposure, for example, upregulates lactate, which is the predominant physiological form of lactic acid, a natural moisturizing factor [2]. Along with the abiotic environmental factors, the microbiome influences the chemical profile of the skin, e.g., through hydrolysis of chemicals secreted by the host or through secretions of secondary metabolites against pathogenic strains such as *Staphylococcus aureus* [5,6]. These pathogenic strains have a role in the development of skin disease, and consequently a diseased skin can be identified by changes in the metabolome, as reviewed in Yan et al., 2017 [7]. Metabolites can furthermore indicate the progression of wound healing in the skin, where linolenic acid is initially increased followed by an increase in adenosine [8]. In summary, taken the recent literature, the skin metabolome gives insights into the physiology of wound healing, topical diseases and their link to systemic comorbidities, and cosmetic aspects and aging related topics.

However, due to the lack of convenient sampling methods, the analysis of skin metabolome has not yet become widespread. A few studies have focused on the metabolome and key metabolites in special applications, including psoriasis, UV-B and nutrition effects, acne, and aging [3,9,10]. They have highlighted the importance of metabolomics to gain deeper knowledge of the pathophysiology of interest. Following the rapid development in this perspective, an increasing demand for studies identifying and cataloging the metabolite composition of the skin will require easy and reproducible sampling technologies and standardization to allow for increased repeatability and better cross-sectional studies.

Currently, there are numerous methods for sampling involving varying levels of invasiveness. For long-term skin secretion studies, hydrogel micropatches are preferred [11,12]. They are made of a network of hydrophilic polymers that “swell” in relation to the amount of sweat absorbed, and have a minimum sampling time of 10 min [11]. For studies focused instead on epidermal compositions, biopsies, suction blisters, tape stripping, and pre-wet swabs have been implemented [2,3,5,13].

Among the less invasive methods, tape stripping and swab analysis are the most widely used. However, tape stripping is improbable for use in skin disease studies [13], as it is painful for skin-sensitive groups and unfeasible for sampling lesions because it removes an already fragile barrier. In consequence, and to ensure patients’ convenience, the collection of skin swabs is the most favorable method. Unfortunately, swab preparation is time intensive and significantly complicates clinical assays [12]. Therefore, here, we introduce an easy and fast sampling method of the skin metabolome and skin secretome, which avoids skin irradiation and is applicable on lesions because it is pain free. We evaluate the collected metabolome systematically and with a focus on recently highlighted key metabolites of skin relevance.

## 2. Results

The skin metabolome was obtained from 22 healthy individuals at the left and right antecubital fossa by swab and WET PREP sampling. Metabolites were analyzed using UPLC-MS^2^, using two complementary separation modes, hydrophilic interaction chromatography (HILIC) and C18 reversed phase (RP), as these are the two most used separation modes in skin metabolomics. Metabolites were assigned to known molecular formulas listed in KEGG (Kyoto Encyclopedia of Genes and Genomes), HMDB (Human Metabolome Database), and LMSD (LIPID MAPS Structure Database) within the absolute error range of 0.005 Da (Appendix A). For both sampling methods, we were interested in the detected skin metabolome in general, as obtained in most individuals (shared skin metabolome), as well as the suitability to obtain inter-individual differences. Furthermore, we analyzed the specific coverage of metabolites that have previously been shown to be important in skin related publications.

First, we focused on the general coverage of metabolome. In HILIC mode, both sampling methods delivered a comparable number of detected metabolites, whereas in RP, swabs outperformed WET PREPs by 920 annotated compounds (Figure 1A). A closer look at the annotation of unique compounds in swabs shows that 55% of them do at least have one annotation as lipid, and between the two columns, RP detects overall 11% more lipids than HILIC (Figure 1B). Apart from this higher lipid measurement in RP, we observed of both separation modes a high percentage (66% in HILIC, 54% in RP) of commonalities in the composition of the metabolome (including the lipidome) in swabs and WET PREPs, as shown in the Venn diagram (Figure 1B). This is further supported by unsupervised statistics, such as principal component (PCA), where the separation of both methods is explained by a small amount of variance in the first two components (Figure 1C). Nevertheless, both PCA and cluster analysis illustrated a distinct separation between the sampling methods, indicating unique detection and quantities of metabolites in each method (Figure 1C and Figure 2).

The shared metabolome, which are metabolites detected stably in most individuals, differed according to method, as observed of the distribution plot of detected metabolites across the samples (Figure 1D). For example, in HILIC mode, 40 of the 44 samples shared 20% of the metabolites found in WET PREPs, while the shared metabolome for swabs was only 10% (Figure 1D). This is substantiated by the PCA (Figure 1C) as a tight clustering of samples illustrates a strong shared metabolome. Additionally, inter-individual variations are reflected in the PCA score plot, and are easily visualized by hierarchical clustering (Figure 2). In consensus with Bouslimani et al., there was a stable metabolome between intra-individual replicates, with only three exceptions for swab samples (Figure 2) [5].

The hierarchical clustering shows a distinct separation according to sampling method for RP and HILIC. Within the swab and WET PREP clusters, two further subclusters are seen that can be explained by including the metadata. To begin with the alignment of the individuals’ metadata, participants resided in one of two urban towns that were 80 km apart, with location B being more densely populated and industrial. Despite the proximity of the locations, the samples clustered based on location for RP swabs, WET PREPs, and HILIC swabs, but in HILIC WET PREPs, location appears as secondary in importance (Figure 2). Apart from a short difference in storage time (less than one week, −80 °C), the sample preparation and analysis were done in a mixed randomized order, and the same person collected the samples in both locations; therefore, extrinsic factors, such as pollution, industry, and population size may be responsible for the clustering. Interestingly, in HILIC WET PREP, the first cluster within the same sampling method was not with respect to location of the sampling, but rather in accordance with peculiarities. For example, participant three did not follow the instructions and showered within 24 h prior, and grouped with other individuals who had showered or had additional peculiarities, such as pregnancy and history of skin disorder (Figure 2). This clustering based on those three peculiarities was also seen in the RP WET PREP samples, but less consistent for the swabs, in both modes RP and HILIC (Figure 2).

For both RP and HILIC, regardless of sampling method, individuals who exercised prior, such as riding a bike or hiking, clustered with the other individual who had no peculiarity, suggesting exercise has a low influence on the skin metabolome (Figure 2). In regard to the biological sex of the participants, there was no overarching grouping that differed to what is seen in sweat metabolomics [14]. Despite the outperformance of WET PREPs in the detection of a stable shared metabolome over swabs, individual differences were also successfully represented in the WET PREP data. Overall, for both sampling methods, there is a stable shared metabolome, but there are also metabolic inter-individual differences, and based on metabolome similarity, bi-lateral replicates are true replicates.

Before we discuss skin relevant metabolites, we would like to discuss differences of the sampling method in regard to metabolic pathways. For this purpose, we performed a Wilcoxon–Mann–Whitney test between swabs and WET PREPs. All *p*-values were Benjamini–Hochberg corrected. Significant metabolites (*p* < 0.05) were then uploaded into MetaboAnalyst [15,16], and all analysis was performed with the focus of impact on the downstream nodes, i.e., out of degree centrality. For the HILIC data, the most impacted and significantly different pathways were aminoacyl-tRNA biosynthesis, glycine serine and threonine metabolism, and valine, leucine and isoleucine biosynthesis. Aminoacyl-tRNA biosynthesis was better represented by WET PREP, while glycine serine and threonine metabolism were better represented in swabs. (Figure 3B). For the RP data, the most impacted and different pathways were aminoacyl-tRNA biosynthesis, valine, leucine and isoleucine biosynthesis, and phenylalanine metabolism, where all three pathways were better represented by WET PREP (Figure 3A). Overall, for both HILIC and RP, we see a general difference in pathways involved with amino acid metabolism, where sampling method mostly affects which amino acids are isolated (Figure 3). This leads us to conclude that sampling method does influence the pathway coverage of the skin metabolome.

Finally, through cross-reference to previous literature, we focus on specific metabolites that have potential relevance to skin function and health (Table 1). These metabolites covered amino acids, their derivatives, acid, sugar, nucleo(t/s)ides, and aromatics. 19 and 18 of these metabolites were detected in HILIC and RP, respectively, with the majority of these metabolites focused on amino acids and derivatives. Out of these, 63% were detected with a higher intensity in WET PREP as compared to swabs, with glutamine, glutamic acid, and ornithine surpassing 0.3 log fold change (Table 1). In regard to sugar and aromatics, caffeine and glucose were significantly higher detected in WET PREP samples, and fucose was better detected by swab. In addition, O-cresol was detected and was significantly higher in WET PREP samples surpassing one log fold change. Finally, the acid retinoic acid was best detected in swabs. In sum, this indicates an advantage of WET PREP to swabs for sampling these skin relevant metabolites, with an outperformance of detection in WET PREPs in comparison to swabs for 75% of compounds in RP and 60% in HILIC.

## 3. Discussion

Metabolomics of the skin has become of increasing interest and clinical importance, yet there is a lack of standard procedures, and, most importantly, there are no studies on the influence of sampling method. There are few methods for skin metabolome collection, and, in consideration that medical studies occur on fragile, weakened skin, we focused on a fast, easy, and, most importantly, painless and non-invasive sampling technique. We compared it with swab sampling, because swabs are the current non-invasive skin sampling standard.

We collected replicates from 22 healthy participants at the left and right antecubital fossa by both WET PREP and swab. These samples were run using two different separation methods (HILIC and RP) in positive electrospray mode. We observed of both methods wide similarities in compound coverage (>55% for both modes RP and HILIC), and, particularly for HILIC, WET PREP has shown to be comparable to swabs considering the total number of annotated compounds. Importantly, WET PREP provides a strikingly higher shared metabolome—defined by metabolites detected stably across samples—in RP and HILIC, while still maintaining unique individual features and more consistent intra-individual replicability, and, for skin relevant metabolites, the collection of compounds by WET PREP was better. In addition, when considering the preparation time and gentleness, WET PREP should be the preferred method for skin metabolomics.

In contrast, when the focus of study is on lipids, we recommend swab as the sampling method. Here, we observed of RP that swabs provide a higher number of compounds, and that lipids make up 55% of them. This is reasonable when considering the sampling procedures. Swabs are pre-wetted with ethanol, a lipophilic solvent, which facilitates the extractions of lipids and the scratching movement of the swab along the skin helps to shear the lipid bilayer. In contrast, in WET PREP, the skin’s surface is simply bathed with water, which naturally hinders lipid extraction.

The shared metabolome may be relevant in future studies to define metabolites for normalization, e.g., creatinine for urine. Here, we would like to stress that the shared skin metabolome is larger in WET PREP samples for both RP and HILIC modes (Appendix A), suggesting better repeatability and stability among individuals when comparing groups. This enhanced repeatability in WET PREP has also been seen within cluster analysis, where intra-individual replicates always clustered together, and this is in accordance with the recent literature [5]. However, the stability also does not sacrifice the individual variability associated with location and individual peculiarities. Beyond just the separation of metabolomes by method, there was an additional clustering factor based on residence of the participants, and future surveys, such as the Consortium of Metabolomics Studies (COMETS), should account for the effect of location in their samples. Even though both our sampling locations are in the south of Bavaria (Germany), intrinsic factors such as environmental pollution and air quality, industrial surrounding, and even population might influence the skin metabolome. We recruited participants from two cities that differ in size. Location B is almost five times as big as location A, and is, in general, more industrialized. In addition, there are differences between locations in air quality [17]. Which environmental factor led in the clustering of our data is beyond the scope of our study, but we would like to highlight the fact that a larger multi-centered study should be performed to determine the potential correlation between the exposome and the skin metabolome. In addition, the peculiarities noted in this data, such as showering within 24 h prior, pregnancy, and a history of dermatitis, did have a significant effect on skin metabolome. This demonstrates that environmental factors affect skin metabolomics, and it highlights the need for full-background questionnaires in skin studies. Exercise did not seem to have an effect, showing that sweat may not be a significant factor. Contrary to the implications of a previous sweat metabolomics study [14], biological sex did not affect the skin metabolome. These differences in results show a need for comparative studies on the metabolites isolated through sweat and dry skin, and suggest sex-based hormones and hormonal studies are best isolated through and performed on sweat.

WET PREP is also preferable when considering metabolites that are influential to the skin. Skin health is maintained through a balance of physiological factors and highly influenced by a variety of environmental factors. Skin relevant small molecules were tested for their detection in swabs and WET PREPs. Overall, we verified the presence of 19 in HILIC and 18 in RP of these metabolites using analytical standard substances, with the majority being collected by WET PREPs, 60% and 75%, respectively. Among them are natural moisturizing factors (NMFs), which are humectants composed of amino acids, their derivatives, and salts [18]. They are important in maintaining both skin hydration and barrier function. 63% of the NMFs are significantly higher in WET PREP samples, with the highest log fold change in glutamine, glutamic acid, and ornithine, an arginine derivative. Interestingly, these three metabolites are all involved in the synthesis of ornithine and arginine, which not only have a role in skin hydration, but also in the formation of malignant skin tumors [19]. This suggests that WET PREP may be the preferential method when researching the metabolomics of skin cancer.

Skin hydration and nutrition are not the only factors that are important to skin health. The skin is dynamically changed by the environment, and bacterial secretion lactate was found on the skin. Lactate has a potential role as a pH skin buffer [20,21], and the collection was comparable for both methods, suggesting that both sampling methods could be used for studies on pH-associated skin diseases, such as atopic eczema [22]. However, when considering the fragility of the skin, we would again recommend the application of WET PREP. Furthermore, lactate is related to microbe–microbe communication [23,24], and both methods show potential for integrating the microbiome with the metabolome. In addition, lactate reflects the effects of UV—a known contributor to skin aging—on the skin [2,25,26].

## 4. Materials and Methods

### 4.1. Participants

We performed two independent sampling collections, first at our clinical location in Augsburg, and second at our partner institution in Munich. For this study, 22 healthy participants with ages ranging from 20 to 50 were recruited. There was an even coverage for both sexes, with 11 men sampled and 11 women. As previous studies have shown that the skin metabolome is influenced by the application of skin hygiene products [5], all volunteers were instructed not to use such products or shower 24 h beforehand, with 10 people having possible extraneous factors—heavy exercise prior, not showering, etc. All collection was consented and ethically approved by the Ethics Commission in the Faculty of Medicine at Technical University Munich (protocol code: 44/16 S on 2 February 2016).

### 4.2. Sample Collection and Processing

Skin samples were collected by WET PREP, and swabs were collected from the left and right antecubital fossae as intra-individual replicates. WET PREP consisted of a total of 5 mL collected in 1 mL increments with sterile water, a plastic ring with inner diameter of 28 mm, and a glass rod. More specifically, a plastic ring was tightly placed against the skin, and 1 mL water added, after which a glass rod was moved in order to release the metabolites from the skin. The water was then collected and placed in a 15 mL tube. This was repeated 5 times in adjacent locations, to make an Olympic ring pattern on the arm. The ease of WET PREP sampling was tested a priori, and we observed that 5 mL could be easily collected and provided a great range of the metabolites. Following collection, the WET PREP was centrifuged at 3000 rpm for 10 min. The swabs were prepared a week in advance to pre-wet and clean for possible contaminations with a 50/50 solution of sterile water listed above, and ultra-pure ethanol where the solution was changed every alternate day [27]. The swab sampling was standardized by length, size, and number of strokes. After sampling, each swab was stored in fresh 500 μL ethanol water solution for 2 h at 4 °C, and vortexed after incubation. The liquid was then filtered, 0.22 μm, to remove bacterial contaminates. We used pure solvents as blanks. All blanks, both WET PREP and swab, were exposed to the air in parallel to sampling. All WET PREP and swab samples were snap frozen in liquid nitrogen, and stored at −80 °C.

Prior to the run with ultra-performance liquid chromatography coupled to mass spectrometry (UPLC-MS^2^), the samples were evaporated in a SpeedVac vacuum concentrator and re-suspended in a mobile phase A. To account for the differences in the sample volume between methods, the WET PREPs were re-suspended with 500 µL and swabs were re-suspended with 50 µL of mobile phase A. Both were vortexed for 30 s. The samples were analyzed in a randomized order. Column equilibration and stability over the entire run was ensured by 10 injections of a quality control mixture consisting of an aliquot of each sample prior to the injection of samples. Additionally, every 10 samples this quality control was injected in order to follow the stability over the entire run. The injection volume was set to 10 µL. The samples were run on data-dependent auto LC-MS^2^ (maXis, Bruker Daltonics, Bremen, Germany, coupled to an UPLC Acquity, Waters, Eschborn, Germany) using both hydrophilic liquid interaction chromatography (HILIC) and reverse phase chromatography (RP) columns being run in both positive electrospray modes, as it had shown better sensitivity in pre-tests for these sample matrices. MS parameters were set following previously described details [28]. We aimed for a fast method, with minimal sample preparation steps to allow for a high throughput. RP run time was from 10 min and for HILIC 12.5 min per sample. RP was performed on a BEH C18 column (100 mm × 2.1 mm ID 1.7 µm, Waters Corporation, Eschborn, Germany) using 0.1% formic acid in water (A) and 0.1% formic acid in acetonitrile (B) as mobile phase. The gradient was set to 5% B for 1.12 min, followed by an increasing proportion of B to 99.5% at minute 6.41 and a plateau for the remaining 3.6 min. Column temperature was kept at 40 °C, and flow rate was set to 0.4 mL/min. For HILIC separation, a iHILIC^®^-Fusion UHPLC column, SS, 100 × 2.1 mm, 1.8 μm, 100 Å column (HILICON AB, Umea, Sweden) was used as previously described [28]. Briefly, mobile phase A consisted of 95% acetonitrile/5% water and 5 mM ammonium acetate, and mobile phase B consisted of 30% acetonitrile/70% water with 25 mM ammonium acetate. The gradient was set to 0.1% B for the first 2 min, followed by an increase from 0.1% to 99.9% B over 7.5 min. 99.9% B was kept for 2.5 min, followed by a fast decrease to 0.1% B within 0.1 min. Flow rate was set to 0.5 mL/min, and column temperature to 40 °C.

Raw data processing was performed in Refiner MS GeneData Expressionist 13.5 (Genedata AG, Basel, Switzerland). Sampling blanks and methodological blanks were subtracted with a two-fold and ten-fold threshold.

### 4.3. Data Analysis/Statistical Evaluation

The *m*/*z* annotation was performed using MassTRIX [29] with a maximum error 0.005 Da, and run against the databases KEGG (Kyoto Encyclopedia of Genes and Genomes), HMDB (Human Metabolome Database), and LMSD (LIPID MAPS Structure Database) without isotopes. No specific organism was chosen in MassTRIX because the skin is highly environmentally influenced by a wide range of microbiota and by the host cells.

R was used for the creation of dendrograms and *t*-tests to determine sampling method specific skin metabolites. Missing values were inputed with a random integer close to the base line limit. Samples were normalized by total sum of intensity and auto-scaled. This was to account for differences in metabolite abundance across samples and allow cross comparisons. For the dendrogram, the distance measurement was spearman with Ward.D2 as the clustering algorithm. A non-parametric *t*-test (Wilcoxon–Mann–Whitney) was used with a false discovery rate of 0.05 and Benjamini–Hochberg (BH) corrected.

Venn diagrams were created in R studio [30]. The package Venn determined the unique metabolites found in each method, and created the cross-individual Venn diagram [31]. The individuals were compared by taking all metabolites, undiscovered and discovered, found in each sample where the metabolite was kept if found in at least one of the bilateral replicates. To create the Venn diagrams with proportional circle size to numerical amount, the package Eulerr was used with the numbers given from Venn [32].

MetaboAnalyst [15,16] was used for the metabolic pathway analysis. Prior to upload, a Wilcoxon–Mann–Whitney test between swabs and WET PREPs was performed with BH correction. Only the significant metabolites (*p* < 0.05) were taken with the first HMDB annotation provided from MassTRIX. All samples were normalized by sum and auto-scaled. Both the intensities and annotation were used for the creation of the pathway analysis chart. The pathway enrichment analysis was performed by a global test and out-degree centrality was determined with the KEGG databased that referenced Homo sapiens.

## 5. Conclusions

Here, we performed metabolomics analysis of 22 individuals using two different non-invasive sampling methods: swab and WET PREP. The results illustrate that the skin has a shared metabolome between participants along with unique individual chemical signatures. Both sampling methods are viable for isolating individual metabolome differences and have wide similarities in compound coverage. However, WET PREP is preferable in terms of intra-individual stability, and provides a larger shared metabolome between participants while still maintaining the individual signatures. WET PREP brings about many possibilities in the field of skin metabolomics and exposomics. It may lead to an increase in biomarker discovery due to its ease of application. If so, WET PREP brings about a bright future for simplicity in skin sampling.

## Figures and Tables

**Figure 1 metabolites-11-00415-f001:**
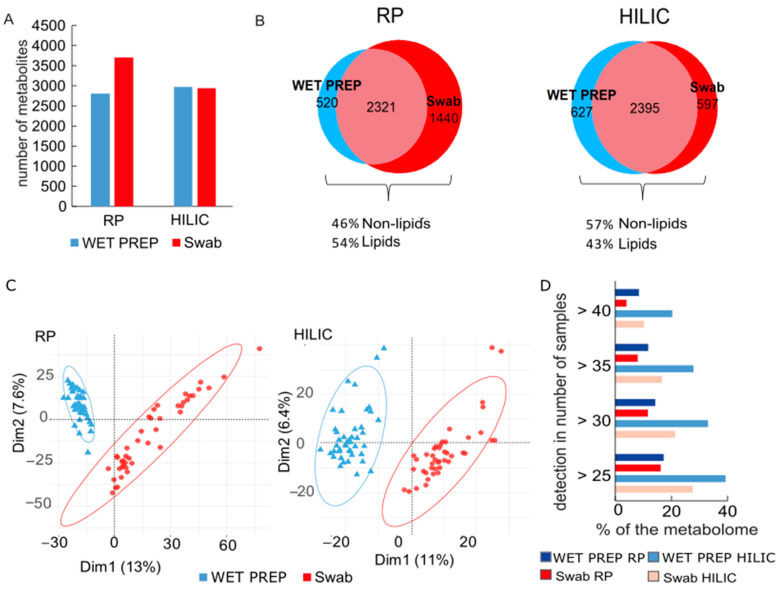
Metabolome overview of skin samples according to sampling method, WET PREP (light blue) and swab (red) for HILIC and RP columns: Total number of annotated metabolites (**A**); Venn diagrams of the total number of compounds according to sampling method with percentage of lipids-compounds had to be present in at least 2 samples per sampling method to be considered (**B**); PCA score plot (**C**); Percentage of shared metabolome across samples with HILIC (lighter colors) and RP (darker colors) (**D**).

**Figure 2 metabolites-11-00415-f002:**
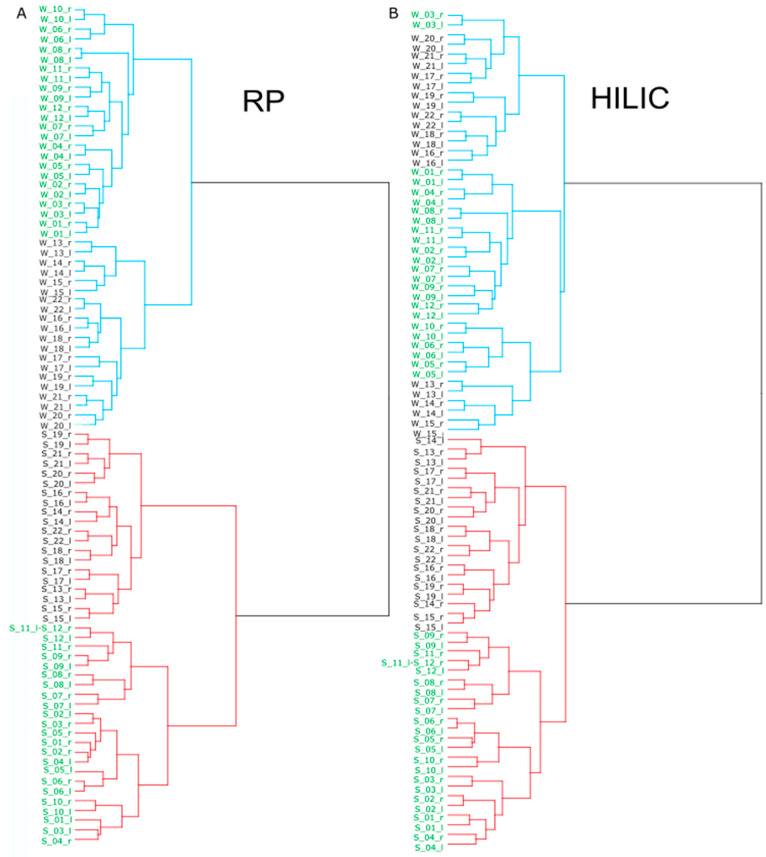
Dendrograms of hierarchical cluster analysis of swab and WET PREP for: RP (**A**) and HILIC (**B**). WET PREP samples noted in blue lines and swabs in red lines. Sample annotation is according to sampling method: WET PREP (W) and swab (S), individual (number following “W” or “S”), lateral side right (r) and left (l), and city of residence, location a (green) and location b (black).

**Figure 3 metabolites-11-00415-f003:**
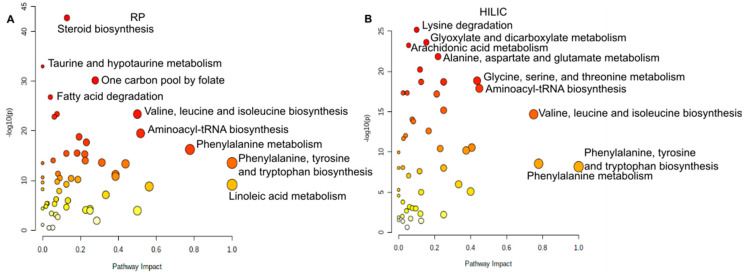
Pathway analysis chart of significantly different metabolites between WET PREP and swab: for RP (**A**) and HILIC (**B**). Metabolites that are significantly different between WET PREP and swab across each column type and with KEGG ID’s, as determined by MassTRIX annotation, were run against the human KEGG database by MetaboAnalyst. The top 20 pathways are shown according *p*-value and impact factor for out of degree centrality shown.

**Table 1 metabolites-11-00415-t001:** Metabolites of skin relevance. Compounds of interest were compared for their detection in swabs and WET PREP samples. Exact *m*/*z* values (0.005 Da) and retention time were used for identification. All compound identification was verified with analytical standards. Log2 fold change WET PREP/swab of the average intensities across all samples are shown. (n.d.) denotes that the compound is not detected. Metabolites are grouped according to chemical type with their references as seen on the far right. (*) denotes a reference from a serum study. Samples were tested for significance by Welch test: *p* < 0.05 (+), *p* < 0.005 (++), *p* < 0.0005 (+++). Scattered detection indicates partial detection in only some of the samples.

Category	Compound	Focus	RP		HILIC		Reference
			Significant Different Detection between WET PREP and Swab	log_2_ Fold Change (Average WET PREP/Average Swab)	Significant Different Detection between WET PREP and Swab	log_2_ Fold Change (Average WET PREP/Average Swab)	
Amino Acid	Taurine	Age	n.d.		+	0.04	Kuehne et al., 2017
	Serine	Psoriasis	+	0.21	+	0.21	Kim et al., 2009
	Proline	Age	+++	0.11	+	0.03	Kuehne et al., 2017
	Threonine	Age	n.d.		++	0.10	Kuehne et al., 2017
	Aspartic acid	Dock8 deficiency	n.d.		n.d.		Jacob et al., 2019 *
	Glutamine	Psoriasis	+++	1.19	+++	only WET PREP	Kim et al., 2009
	Glutamic acid	Psoriasis	+	0.61	−	−0.14	Dutkiewics et al., 2016
	Histidine	Cancer	+++	0.32	n.d.		Taylor et al., 2020
	Phenyl alanine	Psoriasis	+	−0.09	+++	−0.11	Dutkiewics et al., 2016
	Tyrosine	Age	+	0.06	+	−0.07	Kuehne et al., 2017
	Tryptophan	Age	+	−0.09	−	−0.04	Kuehne et al., 2017
Amino Acid Derivative	Hypotaurine	Dock8 deficiency	+++	only WET PREP	n.d.		Jacob et al., 2019 *
	Pyroglutamic acid	Skin	-	0.02	−	0.14	Joo et al., 2012
	Ornithine	Age	+++	0.72	scattered detection		Kuehne et al., 2017
Acid	Lactic acid	Psoriasis	−	−0.08	n.d.		Dutkiewics et al., 2016
	Retinoic acid	Age	+++	−0.22	n.d.		Kuehne et al., 2017
Sugar	Fucose	Age	n.d.		+++	−0.15	Kuehne et al., 2017
	Glucose	Age	scattered detection		+	0.04	Kuehne et al., 2017
Nucleo(t/s)ides	Uracil	Age	−	0.03	+	−0.18	Kuehne et al., 2017
	Guanosine	Atopic Eczema	scattered detection		scattered detection		Jacob et al., 2019 *
Aromatic	Cresol	Age	scattered detection		+++	1.11	Kuehne et al., 2017
	Caffeine	Atopic Eczema	scattered detection		+	0.49	Jacob et al., 2019 *

## Data Availability

All data was deposited into MetaboLights with the accession number https://www.ebi.ac.uk/metabolights/MTBLS2941 (accessed on 14 June 2021).

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
