# Peer review of "Enhanced Access to the Health-Related Skin Metabolome by Fast, Reproducible and Non-Invasive WET PREP Sampling"

_metabolites, 2021, doi:10.3390/metabo11070415_

Round 1

Reviewer 1 Report

The manuscript describes and validates a new methodology for skin metabolome sampling. The authors have compared metabolites recovered with their sampling method with the ones taken from a widespread method (based on rubbing a swab with ethanol) using an LC-HRMS methodology. I think that skin metabolomics will be a hotspot topic, as it’s non-invasive and reveals important information, as skin is a barrier between the organisms and the environment. However, I believe some major revisions are needed to improve the quality of the manuscript.

  1. 49: Change “marco” by “macro”
  2. 56: Put “also” out.
  3. 66: Consider changing “analytical methodology” by “sample treatment”, as in my opinion analytical methodology also refers to instrumental analysis.
  4. 79-81: Consider rewriting this sentence. I find it a bit confusing.
  5. 95 and 96: HILIC and RP are abbreviations of “hydrophilic Interaction liquid chromatography” and “Reversed phase”, so words between parenthesis must be RP and HILIC, not vice versa.

l.108: Lipids were extracted with sampling methods and analysed with RP or with HILIC. Please modify the word extracted.

Figures: Take care about figures naming, as there is no Figure 1. Change Figure names and unify.

Figure 2A (The one with the histogram): In HILIC, blue bar (WET PREP) looks higher than red bar (swab) while the authors have stated that Swab have found 21 compounds more in Swab than in WET PREP (See Venn diagram). Red bar must be higher than blue one.

Figure 2B (The one with Venn diagrams): Are the authors including a compound when they found it at least in one sample?

l.111: PCA means “Principal Component Analysis”, please modify it. In addition, in Figure 2C (Scores Plot of the PCA) first 2 dimensions are explaining only 20.6 and 17.4% of total variance. I think that with these results authors must state in the text that both methods have wide similarities in compound coverage, and PCA is showing that compounds different between both treatments (35% in RP and 23% in HILIC) are causing a small variation in the PCA.    

l.138: I want to suggest the authors an idea. Can these differences be related with contaminants differences in the environment? Are these cities really different in size or industrial production? If so, I can be further studied, as it will add extra value and interest for other experiments (e.g. exposomics).

l.148: Can the authors include a feasible explanation of this grouping? Is it because swab is extracting more compounds and small differences in WET PREP are enhanced by clustering? I think that authors must critically evaluate these differences.

l.162-166: Rewrite these two sentences. They are confusing. In addition, I have a doubt/suggestion: Does differences pointed out in Figure 4 (The one with pathway analysis chart) mean that these metabolites (reflected in pathway differences) are present in WET PREP and absent in swabs (or vice versa), or it only means that there are differences in concentrations? In addition, I think that the authors must discuss which pathway is better represented in which sampling method. Otherwise, the Figure is not understandable, as it only marks differences without giving information of which method is better reflecting each pathway.

l.163: I will find really interesting to include in supplementary material, the list of metabolites “tentatively identified” by MassTrix.

l.183: Is the only origin of glucose in skin the diet? In addition, retinoic acid is not only a cosmetic supplement, but a vitamin metabolite which naturally occur in skin.

l.186: I think that conclude superiority of WET PREP with a selection of 19 compounds is too optimistic. Maybe authors must be cautious in their asseverations, also observing that swab shows 1143 compounds which are not present in WET PREP extract (27% of detected features), despite being lipids the 82% of them (so the other 18% (206 compounds) are not detected with WET PREP). In addition, in metabolomics relevant metabolites are not predefined, so I find that this sentence should be reconsidered.

l.206: “provides a strikingly higher shared metabolome” is senseless, as shared means that it is equal in two sites, so it cannot be superior. If the authors refer to the areas observed in them, please provide data to support this sentence.

l.207: Why the authors know that WET PREP has shown more consistent intra-individual replicability? The authors have only demonstrated that a 0.2% of features are observed in more than 25 individuals in swab, and 0.4% in WET PREP, being a 0.05% more (approx.) in more than 40 individuals. In addition, they do not give the identities of these compounds, so it’s difficult to evaluate if the presence of these compounds reflects a consistent intra-individual replicability or it is related to other sources (e.g. exogenous metabolites, xenobiotics, etc.).

l.234: Change “presents” by “presence”.

l.293: I do not fully understand how WET PREP method is performed. Do it consist in putting 5 mL of sterile water (5 times with 1 mL) directly in the skin and then recover it? Why did the authors recover it? Please consider giving a better explanation (like a protocol) on how to perform WET PREP sampling for readers who have never used it and want to replicate it. In addition, have the authors considered the surface of skin sampled? The authors have resuspended samples regarding the initial volume of liquid, (l. 305-307) but depending on the surface considered for WET PREP method, concentration extracted with both methods can differ from the volume of liquid they have used, being more related with the surface of the skin sampled.

l.308: For column equilibration you need to inject QC sample before randomized batch, to ensure that column conditions are equal for all the samples. Is it what the authors have done? If so, please rewrite the sentence.

l.315: Change “descried” by “described”

l.326-328: Gradient description is a bit strange in my opinion. Consider starting with the first 5 min at 1%, then the increase and finally the 99% B.

l.329: Please, describe how you have taken sample blanks and methodological blanks.

Table 1 caption: IN “(n.d.) denote that the sample is not detected”, change sample by compound.

Supp. Table: There are only 22 participants in this table, while authors have included 24 in the experiment. Please add the 2 participants not included in this table.

Reviewer 2 Report

The submitted manuscript presents a novel rapid, simple, and painless, non-invasive sampling technique (i.e., WET PREP). The method is suitable for metabolomics research in clinical studies of fragile or weakened skin. The authors implemented both RP and HILIC LC-MS (i.e., positive ESI) to evaluate the novel method, which provides a comprehensive metabolomics evaluation.

This is a very important method development for metabolomics research that focus on skin analysis. With the exception of lipid profiles, the novel method (i.e., WET PREP) showed comparable results with the traditional method (i.e., swab). However, the presented method is interesting. 

The authors discussed the variation in skin metabolome as a result of some factors such as showring, exercise and other activities. In addition, instrumental and sample preparation drift can cause a biased result. The authors have applied total area normalization followed by scaling. Have the authors thought of new potential method for skin metabolome normalization?  for example, urine metabolomics data are normalized using creatinine, pH, osmomolity etc. Is there a unique marker that could be used for skin metabolomics normalization? 

Why did authors select positive MS analysis in this study? it is not clear in the manuscript? 

Round 2

Reviewer 1 Report

Now I think the manuscript readability and general quality have increased a lot. The most striking parts have been correctly assessed, so I think it is ready for publication.